# Chemical Profiling and Biological Properties of Essential Oils of *Lavandula stoechas* L. Collected from Three Moroccan Sites: In Vitro and In Silico Investigations

**DOI:** 10.3390/plants12061413

**Published:** 2023-03-22

**Authors:** Taoufiq Benali, Ahmed Lemhadri, Kaoutar Harboul, Houda Chtibi, Abdelmajid Khabbach, Si Mohamed Jadouali, Luisa Quesada-Romero, Said Louahlia, Khalil Hammani, Adib Ghaleb, Learn-Han Lee, Abdelhakim Bouyahya, Marius Emil Rusu, Mohamed Akhazzane

**Affiliations:** 1Environment and Health Team, Polydisciplinary Faculty of Safi, Cadi Ayyad University, Marrakech 46030, Morocco; 2Laboratory of Natural Resources and Environment, Polydisciplinary Faculty of Taza, Sidi Mohamed Ben Abdellah University of Fez, B.P. 1223 Taza-Gare, Taza 30050, Morocco; 3Laboratory of Biotechnology, Conservation and Valorisation of Natural Resources (BCVRN), Faculty of Sciences Dhar El Mahraz, Sidi Mohamed Ben Abdellah University, B.P. 1796, Fez 30003, Morocco; 4Department of Biotechnology and Analysis EST Khenifra, Sultan Moulay Sliman University, Khenifra 23000, Morocco; 5Escuela de Nutrición y Dietética, Facultad de Ciencias Para el Cuidado de la Salud, Universidad San Sebastián, General Lagos 1163, Valdivia 5090000, Chile; 6Laboratory of Analytical and Molecular Chemistry, Multidisciplinary Faculty of Safi, Cadi Ayyad University, Safi 46030, Morocco; 7Novel Bacteria and Drug Discovery Research Group (NBDD), Microbiome and Bioresource Research Strength (MBRS), Jeffrey Cheah School of Medicine and Health Sciences, Monash University Malaysia, Bandar Sunway, Subang Jaya 47500, Malaysia; 8Laboratory of Human Pathologies Biology, Department of Biology, Faculty of Sciences, Mohammed V University, Rabat 10100, Morocco; 9Department of Pharmaceutical Technology and Biopharmaceutics, Faculty of Pharmacy, Iuliu Hatieganu University of Medicine and Pharmacy, 400012 Cluj-Napoca, Romania; 10Engineering Laboratory of Organometallic and Molecular Materials and Environment, Faculty of Sciences Dhar El Mahraz, Sidi Mohamed Ben Abdellah University, Fez 30000, Morocco

**Keywords:** *Lavandula stoechas*, antioxidant, antibacterial, SARS-CoV-2, in silico, GC-MS-MS analysis, medicinal plants, docking

## Abstract

The aim of this study was the determination of the chemical compounds of *Lavandula stoechas* essential oil from Aknol (LSEO_A_), Khenifra (LSEO_K_), and Beni Mellal (LSEO_B_), and *the* in vitro investigation of their antibacterial, anticandidal, and antioxidant effects, and in silico anti-SARS-CoV-2 activity. The chemical profile of LSEO was determined using GC-MS-MS analysis, the results of which showed a qualitative and quantitative variation in the chemical composition of volatile compounds including L-fenchone, cubebol, camphor, bornyl acetate, and τ-muurolol; indicating that the biosynthesis of essential oils of *Lavandula stoechas* (LSEO) varied depending on the site of growth. The antioxidant activity was evaluated using the ABTS and FRAP methods, our results showed that this tested oil is endowed with an ABTS inhibitory effect and an important reducing power which varies between 4.82 ± 1.52 and 15.73 ± 3.26 mg EAA/g extract. The results of antibacterial activity of LSEO_A_, LSEO_K_ and LSEO_B_, tested against Gram-positive and Gram-negative bacteria, revealed that *B. subtilis* (20.66 ± 1.15–25 ± 4.35 mm), *P. mirabilis* (18.66 ± 1.15–18.66 ± 1.15 mm), and *P. aeruginosa* (13.33 ± 1.15–19 ± 1.00 mm) are the most susceptible strains to LSEO_A_, LSEO_K_ and LSEO_B_ of which LSEO_B_ exhibits bactericidal effect against *P. mirabilis*. furthermore The LSEO exhibited varying degrees of anticandidal activity with an inhibition zones of 25.33 ± 0.5, 22.66 ± 2.51, and 19 ± 1 mm for LSEO_K_, LSEO_B_, and LSEO_A_, respectively. Additionally, the in silico molecular docking process, performed using Chimera Vina and Surflex-Dock programs, indicated that LSEO could inhibit SARS-CoV-2. These important biological properties of LSEO qualify this plant as an interesting source of natural bioactive compounds with medicinal actions.

## 1. Introduction

Medicinal plants are of great interest as a source of bioactive molecules used to treat different human diseases [1]. Among the secondary metabolites, essential oils (EOs) have applications in the pharmaceutical and aromatic industries. It is for this reason that their use has increased during the last decade directly or indirectly in daily life [2]. Several studies have suggested the employment of EOs instead of synthetic chemicals in the treatment of human pathologies. The biosynthesis of bioactive products, the contents in EOs, and their biological effectiveness may vary based on many parameters, such as geographical variation, matrix use, phenological stages, seasonal variation, light availability, interaction, and anthropogenic activity [3,4,5,6,7,8,9]. Considering these factors, research is focused on identifying the optimal conditions to obtain EOs with rich content of bioactive molecules.

*Lavandula stoechas* L., Lamiaceae family is one of the 39 species in the *Lavandula* genus, and it is widely used throughout the Mediterranean region for its medicinal interests attributed to its bioactive compounds, including camphor, myrtenol, erythrodiol, lupeol, terpineol, eucalyptol, fenchone, luteolin, oleanolic acid, pinocarvyl acetate, and ursolic acid [10]. Lavender, due to its phytochemical composition, is a popular medicinal and aromatic plant commonly used in traditional medicine and food and cosmetic industries thanks to its key antioxidant, anti-inflammatory, and antimicrobial properties. According to ethnobotanical and ethnopharmacological investigations, *L. stoechas* is used in Morocco to treat inflammatory problems, nephrotic syndromes, and rheumatic diseases, as an antispasmodic agent, and to reduce pain. In Portugal, the aerial part is used to treat heartburn and sea-sickness and to enhance blood circulation [11,12]. In Turkey and Spain, it is used by women to regulate menstrual cycles as a carminative and antispasmodic [13,14]. The plants can also be used as an antidiabetic, to relieve kidney stones, and in the treatment of hypertension, epilepsy, migraine, and otitis [15,16]. From the point of view of the pharmacological activities of LSEO, several research works have evaluated their antimicrobial, antioxidant, antileishmanial, insecticidal, and anticancer activities [17,18,19,20,21,22,23]. These biological properties may be attributed to the high content of the LSEO fenchone/camphor chemotype. However, the results of the biological activities differ from one study to another, whose differences could be due to the quantitative and qualitative variation of the essential oil chemical composition [10], which might be influenced by the parameters mentioned above.

To the best of our knowledge, no study has been completed to report on the chemical profiles or biological activities of LSEO extracted from plants from different regions in Morocco. Therefore, the aim of this work was to evaluate, in vitro, the antioxidant, antibacterial, and anticandidal effects, as well as the anti-severe acute respiratory syndrome coronavirus 2 (SARS-CoV-2) action in silico of LSEO collected from three different Moroccan sites, Taza city (Northern Morocco), Khenifra, and Beni Mellal (Central Morocco), and thus find the optimal site to collect this species for use in alternative medicine or as a potential therapy in conventional medicine.

## 2. Materials and Methods

### 2.1. Collection of Plants and Isolation of Essential Oils

Plant samples were collected in April 2021 from three sites Sebt Malal Aknol, Aguelmous, and Moujd located in the provinces of Taza, Khenifra, and Beni Mellal, respectively. The identification of plants was achieved by Pr. Abdelmajid Khabbach in the Natural Resources and Environment Laboratory of the Polydisciplinary Faculty of Taza, Sidi Mohamed Ben Abdellah University of Fez. The dried leaves (100 g) were subjected to hydrodistillation using a Clevenger type apparatus for 3 h. The essential oil was stored at 4 °C until use.

### 2.2. GC-MS-MS Analysis of LSEO

The chemical composition of LSEO collected from the mentioned three sites was analyzed using GC-MS-MS analysis [24]. The investigation was performed on gas chromatography TQ8040 NX (Shimadzu, Tokyo, Japan) attached to a triple quadrupole, tandem mass spectrometer (GC-MS). Chromatography was conducted on an apolar, equipped with capillary column RTxi-5 Sil MS column (30 m × 0.25 mm ID × 0.25 µm). Purified helium was used as carrier gas, and the injection volume was 1 µL. Temperature of the source was 200 °C. The chromatographic system was programmed with splitless injection (split opening at 4 min), an injector temperature of 250 °C, and pressure of 37.1 kPa. Temperature was programmed with an initial temperature of 50 °C for 2 min, ramp 1 and ramp 2 were 5 °C/min to 160 °C for 2 min, and 5 °C/min to 280 °C for 2 min, respectively. The identification of each compound was based on its mass spectra (MS) and by computer matching with standard reference databases.

### 2.3. Antioxidant Activities

#### 2.3.1. Free Radical Scavenging Activity by ABTS^+^

The radical scavenging activity of LSEO against the radical ABTS^+^ was evaluated according to the Brahmi et al. [25], with some modifications. First, the ABTS^+^ solution was prepared at 7 mM concentration with potassium persulfate (2.45 mM); this solution was allowed in obscurity at room temperature for 12 h. Before tests, the ABTS^+^ stock solution was diluted with methanol to an absorbance of 0.700 ± 0.020 at 734 nm. Then, 75 µL of test samples at different concentrations (31.12–500 µg/mL, prepared in methanol) were added to 925 µL of ABTS solution. The absorbance was measured at 734 nm using a spectrophotometer (SPECUVIS1, UV-Visible). Ascorbic acid was used as standard antioxidant.

The antioxidant activity (AA) was calculated using the following Formula (1):AA (%) = (Abs_control_ − Abs_sample_)/Abs_control_ × 100(1)
where Abs_control_ is the absorbance of the negative control, and Abs_sample_ is absorbance of the test sample.

#### 2.3.2. Reducing Power Assay

The reducing power activity (FRAP) of LSEO was evaluated according to our previous study [26]. Indeed, the solution made up of the phosphate buffer (2.5 mL, 0.2 M, pH 6.6), potassium ferricyanide (2.5 mL), and the test samples (1 mL at 1 mg/mL dissolved in methanol) was prepared. To stop the reaction, trichloroacetic acid (10%) was added at a volume of 2.5 mL after incubation for 20 min at 50 °C (water bath). Then, the mixture was centrifuged at 3000 rpm/min for 10 min. Afterward, 2.5 mL supernatant was mixed with 0.5 mL of 0.1% ferric chloride and 2.5 mL of distilled water. Finally, absorbance was measured at 700 nm using a spectrophotometer (SPECUVIS1, UV-Visible). The reducing power is expressed in milligram equivalence of ascorbic acid per gram of extract (mg EAA/g).

### 2.4. Antibacterial Activity

#### 2.4.1. Pathogen Bacteria and Growth Conditions

Antibacterial activity was performed against pathogen bacteria, including Gram-positive bacteria (*Bacillus subtilis* DSM 6633 and *Staphylococcus aureus* CECT 976) and Gram-negative bacteria (*Proteus mirabilis* INH, *Escherichia coli* K12, and *Pseudomonas aeruginosa* CECT 118), using the disc diffusion method as described in our previous work [1]. First, sterile disks (6 mm diameter) were applied onto the surface of the MHA, which was previously spread with the test inoculum concentrations, and were loaded with a volume of 12.5 µL of pure essential oil. Gentamicin (15 µg) served as a positive control and 10% dimethylsulfoxide (DMSO) as negative control. After incubation, the antibacterial effect was determined by calculating the diameter of inhibition zones.

#### 2.4.2. Minimum Inhibitory Concentration and Minimum Bactericidal Concentration

The MIC values were evaluated in sterile 96-well microplate according to [27], with some modifications. First, 100 μL of Mueller–Hinton Broth (MHB) was distributed in all test wells except the first well in which a volume of 200 μL was added containing the LSEO with a concentration of 25 mg/mL in 10% DMSO. A series of doses varying from 25 to 0.097 mg/mL were prepared from the first to the ninth well. Then, 10 μL of the suspension from each well was removed and replaced by the inoculum test concentration, except the 10th well, which was used as sterility control. The last two wells (eleventh and twelfth) were considered as positive growth negative controls, which contained only MHB broth and 10% DMSO (*v*/*v*) without LSEO, respectively. Then, the plates were incubated at 37 °C for 24 h. After the incubation, a volume of 25 μL of an indicator of microorganism’s growth was added to each well; 2,3,5-triphenyltetrazolium chloride (TTC) was prepared at a concentration of 5 mg/mL in sterile distilled water. The microplate was re-incubated at 37 °C for 30 min. The minimum bactericidal concentration (MBC) was determined by the inoculation in MHA of 10 μL of broth from the uncolored wells and incubated at 37 °C for 24 h.

### 2.5. Anticandidal Effect

The anticandidal activity of pure LSEO was evaluated against *Candida albicans*, which was cultured in YPGA medium (5 g yeast extract, 5 g peptone, 10 g glucose, and 15–18 g agar in 1 L) and incubated at 37 °C for 48 h. The effect was evaluated using disc diffusion method [28].

### 2.6. Anti-SARS-CoV-2 In Silico

#### 2.6.1. Molecular Docking

Molecular modeling is an interesting in silico tool used to determine the stability of compounds and the interaction types responsible for antiviral biological activity. Different EO studies revealed antiviral activity against SARS-CoV-2 [29,30,31]. Two different software were used: Surflex-Dock and UCSF chimera in UCSF Chimera 1.13.1 [32,33]. The crystal structures were edited to remove water molecules, and all hydrogen atoms were added to the structure. For Surflex-Dock, protomol-specified residues in the protein were applied to determine the docked cavity of the receptors. All ligands were docked using automatic docking method, and total scores were expressed in -log10 (Kd) units to show binding affinities [32]. For UCSF Chimera, the 3D structure of both receptors (PDB:6lu7 and PDB:6vsb) were loaded to chimera window and prepared using Dock Prep mode. Polar hydrogens were added, and Gasteiger charges were calculated. The docking analyses of studied proteins were executed using the plug-in of Chimera Vina. The binding sites were identified using native ligand with a grid box of size 20 × 20 × 20 centered at x = 247.84, y = 255.31, z = 272.31 Å and x = −12.17, y = 13.96, z = 69.74 Å for both receptors PDB:6vsb and PDB:6lu7, respectively [34,35]. The native ligand was deleted before docking, and the conformations were searched with binding parameters of 3 kcal/mol as the maximum energy difference, 8 as exhaustiveness of search, and 9 as the number of binding modes. Root mean square deviation (RMSD) values were used to compare the ligand between the predicted and its corresponding crystal structure [36]. The lowest energy-minimized pose was used for further analysis. Discovery Studio 2016 software was utilized to visualize the different interactions of molecular docking results [37].

#### 2.6.2. ADMET Properties

Pharmacokinetics is an important process that studies drug absorption, distribution, metabolism, excretion, and toxicity (ADMET). It is a fundamental concept to eliminate low drug candidates, which may present problems during in vivo studies, and it also determines the availability of a drug candidate [38]. ADME/T property predictions allow drug developers to understand the safety and efficacy of a drug candidate, as it is necessary for a drug developer to make a go/no-go decision in the late stages of preclinical and clinical programs. In this study, ADMET properties were determined using pkCSM online server [39].

#### 2.6.3. Molecular Prediction

With the aim of determining the potential bioactive compounds that exist in *L. stoechas* plants and finding drug candidates against viral infections, molecular docking has been performed. Molecular docking is used to predict how receptors interact with bioactive compounds (ligands). Several studies investigated the bioactive compounds in plants that have potential to inhibit the proliferation of viruses [31,40,41]. Moreover, a new study reported that an inhibitor of HIV protease (nelfinavir) was predicted to be COVID-19 drug candidate using molecular docking [42].

The compounds docked were molecules found in high percentages in *L. stoechas* plants gathered from the interested regions in Morocco. These compounds were L-fenchone, camphor, bornyl acetate, cubebol, viridiflorol, and tau-muurolol.

Three-dimensional (3D) structures, main protease Mpro and spike glycoprotein targets of SARS-CoV-2, were retrieved from Protein Data Bank [34,35] in pdb formats. These proteins were chosen as receptors in molecular docking process. Water molecules and ligands that were still attached to the receptor were removed. The receptor was stored in the pdb, and polar hydrogen atoms were added. Docking preparations, analyses, and determination of hydrogen bonds (H-bonds) were conducted using two different software, Chimera 1.15 (vina) and sybyl-x 2.0 (Surflex-Doc). The visualization of receptor–ligand interactions was obtained using BIOVIA Discovery Studio Visualizer 2016 [37].

### 2.7. Statistical Analysis

All assays were done in triplicates. Values of each test were expressed as mean ± standard deviation (SD) and were subjected to analysis of variance (one-way ANOVA). The statistical analysis was performed using GraphPad Prism version 6.00 (GraphPad Inc., San Diego, CA, USA). Differences (between groups) were considered as statistically significant at *p* < 0.05.

## 3. Results and Discussion

### 3.1. Chemical Composition

The essential oil yields (*w*/*w*) were 1.84, 0.79, and 0.65% for LSEO_K_, LSEO_A_, and LSEO_B_, respectively. The results of the GC-MS-MS analysis showed the richness of the plants collected from the three regions in volatile compounds with variability between the three essential oils analyzed. Indeed, LSEO_A_ contains L-fenchone (14.39%), Ɣ-1-cadinene aldehyde (10.61%), viridiflorol (8.54 %), bornyl acetate (8.39 %), and myrtenyl acetate (3.77%) as the main compounds or chemotypes (Table 1).

LSEO_B_ showed the presence of cubebol (22.68%), camphor (22.29%), borneol (5.15%), muurol-5-en-4-one <cis-14-nor-> (4.21%), L-fenchone (4.03%), and silphiperfol-5-ene (3.27%) as the main compounds (Table 2).

However, τ-muurolol (18.44%), cubebol (16.07%), camphor (13.39), muurol-5-en-4-one (cis-14-nor-) (6.84), selina-3,7(11)-diene (4.5%), 3-adamantan-1-yl-butan-2-one (4.39%), borneol (3.26%), linalool (3.02%), and benzenemethanol, 4-(1-methylethyl) (3%) were the main compounds in LSEO_K_ (Table 3).

The literature reports supported these findings concerning other medicinal plants. Indeed, several studies have reported the chemical composition of LSEO, with some indicating that, in addition to the fenchone/camphor chemotypes, the chemical compositions of LSEO collected in Morocco and Greece contained 1,8-cineole and camphene, and α-cardinol, respectively, while others disclosed the presence of myrtenyl acetate, bornyl acetate, linalyl acetate, camphene, linalool, borneol, γ-terpinene, lavandulyl acetate, and caryophyllene as major compounds [10,18,43,44,45,46]. Besides the presence of some main compounds, our study clearly revealed the chemical composition quantitative and qualitative variability of *L. stoechas* plants collected from three different regions. This confirms the idea postulating that the environmental, climatic, and nutritional conditions of the same plant impact, quantitatively and qualitatively, the synthesis of secondary metabolites. Several previous works have revealed this causal link between the variation of external factors, such as temperature, humidity, soil, or climate type, metabolic pathways, and the chemical composition of EOs. Therefore, the nature of soil may induce different elicitor production, a group of molecules secreted by microorganisms in soil (at the rhizosphere), which stimulate and regulate the synthesis and accumulation of secondary metabolites in medicinal plants [47]. Moreover, it has been previously shown that environmental factors could change the synthesis of EOs via different epigenetic modifications or the alteration of gene expression involved in secondary metabolite anabolism [48,49].

It was also exposed that LSEO chemical compounds might vary between seasonal stages and plant parts (stems, leaves, and flowers) [50]. Indeed, the findings disclosed that LSEO expressed volatile substances according to phenological stages and plant parts with remarkable variability.

### 3.2. Antioxidant Activity

The antioxidant activities of LSEO were examined using ABTS and FRAP tests. An ABTS radical scavenging assay, based on the transfer of both a hydrogen atom and an electron, measures the capacity of antioxidants to neutralize ABTS, a blue-green stable radical cation, enabling the quantification of the antioxidant capability of both hydrophilic and lipophilic compounds. The results show that LSEO_K_ and LSEO_B_ have a greater capacity to reduce ABTS compared to LSEO_A_ (Figure 1). For a dose of 220 µg/mL, the percentage of inhibition exceeded 90%.

The FRAP test is based on the transfer of one electron and measures the reduction of the ferric ion (Fe3^+^)–ligand to the blue ferrous (Fe2^+^) complex in acidic pH conditions to maintain iron solubility. For this test (Figure 2), LSEO_B_ presented a significant value of 15.73 ± 3.26 mg EAA/g extract, while that of LSEO_A_ and LSEO_K_ were 6.91 ± 0.47, and 4.82 ± 1.52 mg EAA/g extract, respectively. The antioxidant potency of LSEO was previously evaluated, and the results demonstrated that they exert important antioxidant activities [18,21,51,52].

### 3.3. Antibacterial Activity

In vitro tests of the antibacterial effect of LSEO, using the filter paper disc diffusion and the microplate methods against microorganism tests, are summarized in Table 4. The findings revealed a variation in sensitivity between the bacteria tested. Concerning the Gram-positive bacteria, *B. subtilis* was the most sensitive strain to LSEO_A_, LSEO_K_, and LSEO_B_ with a diameter of inhibition zone of 25 ± 4.35, 21.66 ± 2.08, and 20.66 ± 1.15 mm, respectively. Among the Gram-negative bacteria, *P. mirabilis* was significantly inhibited by LSEO_K_ (22.66 ± 0.57 mm) compared to LSEO_B_ and LSEO_A_ with 20 ± 1.00 and 18.66 ± 1.15 mm, respectively. In addition, significant inhibition was exerted by LSEO_K_ against *P. aeruginosa* (19 ± 1.00 mm) in comparison with LSEO_B_ and LSEO_A_ with 15.66 ± 0.57 and 13.33 ± 1.15 mm, respectively. The MBC/MIC values inform that LSEO_A_, LSEO_K_, and LSEO_B_ exert a bacteriostatic effect versus all bacteria tested except LSEO_B_, which exhibits a bactericide effect against *P. mirabilis*. From the point of view of the difference in the antibacterial potential of LSEO_A_, LSEO_K_, and LSEO_B_, our results could be attributed to the qualitative and quantitative variation in their chemical composition with the active compounds.

Concerning the susceptibility of Gram-positive and Gram-negative bacteria, it has been revealed that the Gram-negative bacteria are less sensitive to plant extracts compared to Gram-positive bacteria because Gram-negative bacteria possess double membranes, which protect them versus the antibacterial products [1,53,54,55]. Our findings showed that LSEO_A_, LSEO_K_, and LSEO_B_ were active against both Gram-negative (*P. aeruginosa* and *P. mirabilis*) and Gram-positive (*B. subtilis*) bacteria. These results may be related to the presence of a high content of active compounds with antibacterial potential. Many studies already confirmed that minor components in the EOs could have synergistic antimicrobial activity [56,57].

### 3.4. Anticandidal Effect

The in vitro anticandidal activity of the LSEO was qualitatively confirmed using the diameter of inhibition zones. The LSEO exhibited varying degrees of antifungal activity. The inhibition zones were 25.33 ± 0.5, 22.66 ± 2.51, and 19 ± 1 mm for LSEO_K_, LSEO_B_, and LSEO_A_, respectively.

### 3.5. Anti-SARS-CoV-2 In Silico

Molecular docking study

Essential oils have shown promise as antiviral agents against several pathogenic viruses [58,59]. To gain structural insights and understand the binding mode of molecular structures and protein targets, we applied molecular docking processes that were previously described as an efficient in silico approach [60]. Various experiments revealed that EOs could contribute to preventing the entry of SARS-CoV-2 into the human body and investigated the efficacy of EO compounds in the prevention and treatment of COVID-19 [31,59,61,62,63]. Da Silva et al. used molecular docking analysis to determine the interaction of 171 essential oil components with SARS-CoV-2, showing that the compound with the best-normalized docking score to SARS-CoV-2 Mpro was the sesquiterpene hydrocarbon (E)-β-farnesene [64].

Two of the very well-characterized and promising drug targets are the main protease (Mpro; 3CLpro) and the papain-like protease (PLpro), which play key roles in viral replication and transcription [65]. They have been the main target of many vaccines as antibodies against this protein block the entry of the virus and inhibit viral replication [66]. There have been several molecular docking studies on these targets as well as EOs molecular docking with SARS-CoV-2 proteins [67,68,69,70]. Moreover, commercially available drugs have been confirmed using in silico methods [71,72].

As the chemical compositions of the researched *L. stoechas* plants gathered from the three regions were different and in order to determine the promising antiviral compounds against SARS-CoV-2, the molecular docking process was performed using Chimera Vina and Surflex-Dock programs. The redocking process of co-crystal ligands for both receptors showed low RMSD values (<1.5), which indicated the reliability of the applied docking process.

In silico molecular docking of the studied compounds, L-fenchone, camphor, bornyl acetate, cubebol, viridiflorol, and tau-muurolol, with the main protease Mpro and S-protein targets was applied. The results presented in Table 5 show that bornyl acetate and cubebol compounds have good binding affinities and an interesting scoring compared to chloroquine, a compound that has been known for quite a long time to inhibit the invasion of different viruses in cultured cells in vitro, including SARS-CoV and MERS-CoV [73,74,75].

The molecular docking of each compound showed 10 different poses; the stable one presented in Figure 3 is the structure used for further studies.

The stable pose of bornyl acetate in the Mpro receptor pocket shown in Figure 4 presents the hydrogen bond with SER A:144 residue and pi-alkyl interactions with CYS A:145, and MET A:49 and MET A:165 residues, showing a score of 3.92 (−5.54 with Chimera Vina). Cubebol shows two hydrogen bonds with SER A:144, CYS A:145 residues, pi-alkyl interactions with MET A:49, and HIS A:41 and LEU A:27 residues, with a score of 3.12 (−5.5 using Chimera Vina). The compounds tau-muurolol and camphor are stabilized by the hydrogen bond, with LEU A:141 and GLU A:166 residues, respectively, but the presence of an unfavorable interaction with the SER A:144 residue for the tau-muurolol compound destabilized its inhibition compared to the rest of compounds. L-fenchone and viridiflorol are stabilized with different pi-alkyl interactions.

The molecular docking of compounds with the spike glycoprotein (pdb:6vsb) receptor presented in Figure 5 shows pi-alkyl interactions between bornyl acetate and LEU C:293 and PHE C:58 and PHE C:59 residues, with a score of 3.55 (−5.53 with Chimera Vina). The compound cubebol is stabilized by two hydrogen bond interactions, with ASN C:606 and LYS C:300 residues. In addition, the presence of pi-alkyl interactions increases the stability of this compound in the receptor pocket, with a score of 3.37 (−5.6 with Chimera Vina). L-fenchone is stabilized by pi-alkyl interactions, with a score of 2.54 (−4.8 with UCSF Chimera). The three compounds tau-muurolol, viridiflorol, and camphor show a hydrogen bond interaction with the ARG B:1224 residue for the two first compounds and with the ARG B:1226 residue for camphor.

Based on the energy affinities presented in Table 5 and the molecular interactions described in Figure 4 and Figure 5, cubebol and bornyl acetate are the compounds that show an excellent inhibition to both receptors, the main protease Mpro (pdb:6lu7) and spike glycoprotein (pdb:6vsb) targets. Moreover, the *L*. *stoechas* plants gathered from Khenifra and Beni Mellal show an interesting cubebol percentage. In addition, LSEO_B_ presents bornyl acetate in its composition, indicating that the LSEO_B_ plant could be a promising SARS-CoV-2 inhibitor.

The results of the current in silico molecular docking process, employing the binding affinity and interactions, support the use of LSEO compounds as possible candidate inhibitors in the treatment of COVID-19.

### 3.6. ADMET Predictions

The Lipinski rule is one of the best filters in the virtual screening of bioactive molecules to determine an effective drug in early preclinical development [76]. The values in Table 6, calculated using pkCSM, indicate that cubebol and bornyl acetate have molecular weights under 500, LogP and hydrogen bond donors less than 5, and rotatable bonds and hydrogen bond acceptors less than 10, with a polar surface under 140 Å^2^, all indicating the drug permeability and ability of these two compounds.

Absorption, distribution, metabolism, excretion, and toxicity studies are essential for determining pharmacological properties to discover bioactive compounds with desirable pharmaceutical properties and therefore discuss their drug availability [77]. The calculation of intestinal absorption, and skin and CaCO_2_ permeability indicate that cubebol and bornyl acetate have high CaCO_2_ permeability (CaCO_2_ > 0.9), high intestinal absorption (a compound with values less than 30% are poorly absorbed), and low skin permeability (a compound with values less than −2.5 has low skin permeability) (Table 7). Moreover, the distribution and metabolism results show that both cubebol and bornyl acetate present no inhibition for main cytochrome enzymes, while cubebol reveals that it can be a CYP3A4 substrate, which may be likely metabolized and present drug–drug interactions.

Cubebol exhibited a high steady-state volume of distribution (VDss), >0.45, and was ready to cross the blood–brain barrier (BBB). Bornyl acetate showed medium VDss and was also ready to cross the BBB. Both compounds disclosed no AMES toxicity or hepatotoxicity, with total clearance of 0.88 and 1.03 for cubebol and bornyl acetate, respectively. These results indicate that bornyl acetate revealed the best pharmacokinetic properties compared to cubebol, and it can be considered in further experiments. Similar to our study, Wei et al., who found linalool and linalyl acetate (29.48 and 40.97%, respectively) in lavender, proved that these major LSEO components had no toxicity and were safe to be used as food or medication [78]. A recent study reported that linalyl acetate (39.7%), linalool (33.6%), and terpinen-4-ol (14.9%) were the most abundant lavender EOs and that they possessed antiviral activities against many DNA and RNA viruses [79].

In silico studies and ADMET prediction of the selected LSEO bioactive molecules demonstrated good pharmacokinetic properties. The phytochemical composition and some biological activity outcomes were slightly different compared to other studies and are a confirmation of the originality of our *Lavandula stoechas* research. The results are very promising and could encourage further in vitro and in vivo evaluations of this plant and its LSEO.

## 4. Conclusions

The present work is a detailed description of the chemical composition and biological effects of essential oils extracted from *Lavandula stoechas* harvested from three Moroccan sites. Our results showed that this plant synthesized various volatile compounds, such as L-fenchone, cubebol, camphor, bornyl acetate, and τ-muurolol, with qualitative and quantitative differences depending on their harvest site. The essential oils were in vitro analyzed for their antimicrobial, antioxidant, and anti-SARS-CoV-2 effects. The inhibition reached 81.1% for the antioxidant activity. The antimicrobial tests disclosed that the essential oils were effective against the growth of *B. subtilis*, *P*. *aeroginosa*, and *P. mirabilis*. In addition, LSEO_K_, LSEO_B_, and LSEO_A_ inhibited the growth of *C. albicans*. In silico investigation of the volatile compounds of essential oils against SARS-CoV-2 revealed a strong affinity of these molecules with the targets of this virus. Future studies should focus on determining and/or validating the pharmacokinetic and pharmacodynamic parameters of the essential oils from *Lavandula*, as well as the toxic effects in clinical trials, before any application in the pharmaceutical, cosmetic, or food industries.

## Figures and Tables

**Figure 1 plants-12-01413-f001:**
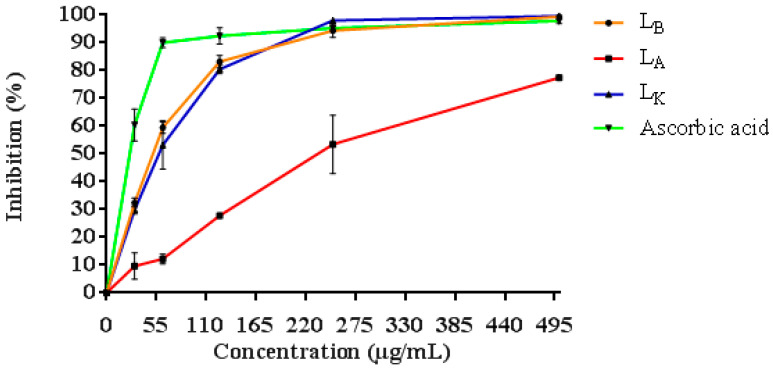
Scavenging activity of LSEO_A_, LSEO_B_, LSEO_K_, and ascorbic acid.

**Figure 2 plants-12-01413-f002:**
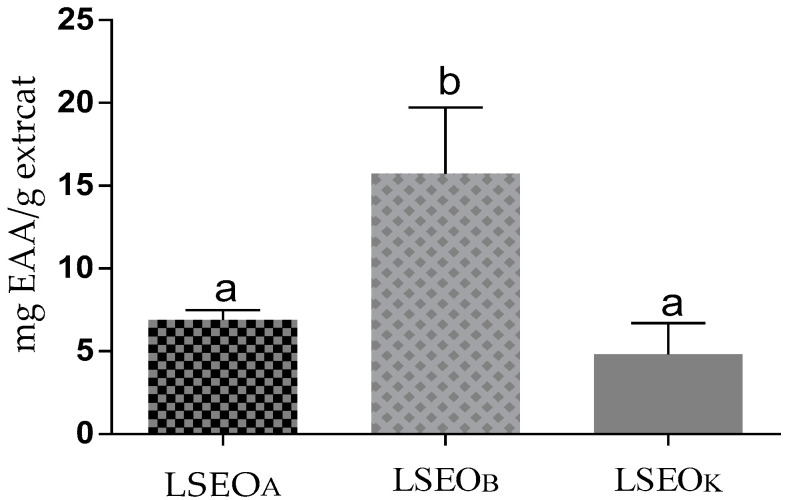
Ferric reducing antioxidant power of LSEO_A_, LSEO_B_, and LSEO_K_ in mg of equivalent ascorbic acid/g of extract (values not sharing a common letter differ significantly at *p* < 0.05).

**Figure 3 plants-12-01413-f003:**
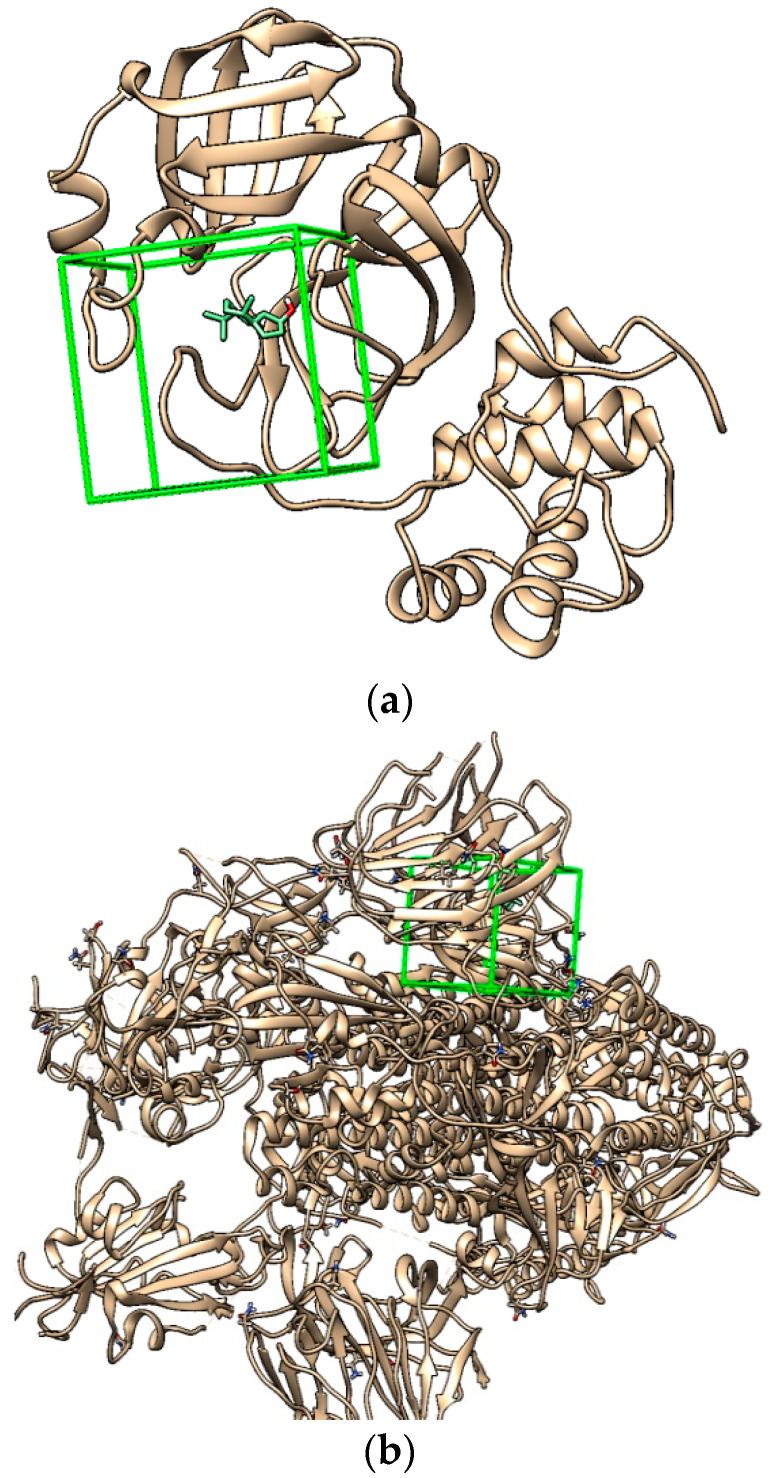
The stable pose of cubebol in receptor pocket using UCSF Chimera: (**a**) main protease Mpro; (**b**) spike glycoprotein targets.

**Figure 4 plants-12-01413-f004:**
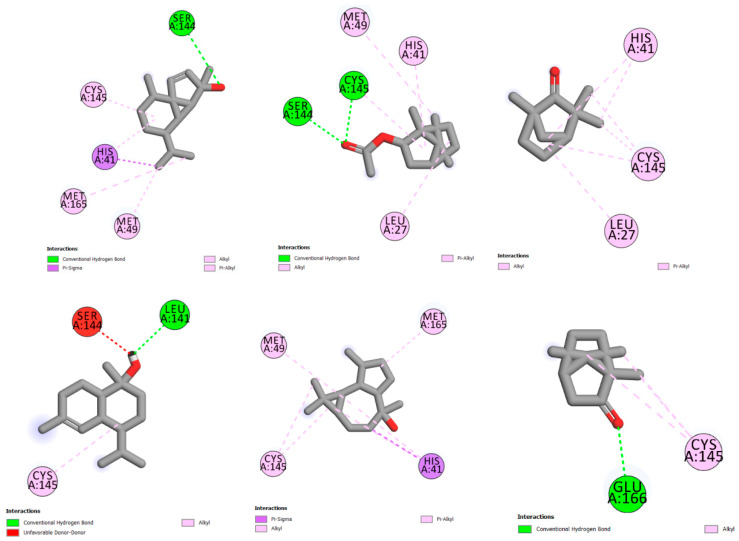
The molecular interactions between the studied compounds and main protease Mpro receptor (pdb:6lu7) using discovery studio visualizer.

**Figure 5 plants-12-01413-f005:**
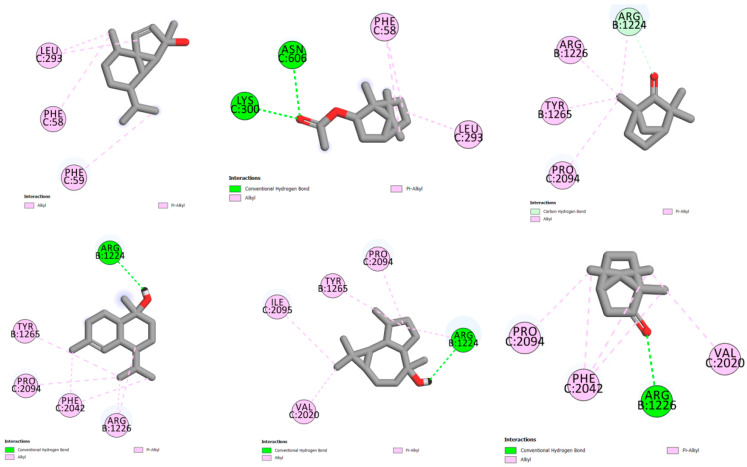
The molecular interactions between the studied compounds and spike glycoprotein receptor (pdb:6vsb) using discovery studio visualizer.

**Table 1 plants-12-01413-t001:** Chemical composition of LSEO_A_.

Peak Number	Compound	Retention Time	Area
1	L-fenchone	12.6	14.39
2	2-norbornanol	13.50	0.98
3	Camphor	14.42	23.80
4	Borneol	15.05	1.13
5	3-adamantan-1-yl-butan-2-one	15.280	1.72
6	Benzenemethanol, 4-(1-methylethyl)	15.47	1.18
7	2-pinen-10-ol	15.75	1.40
8	2-pinen-4-one	16.07	1.02
9	2-cylohexen-1-ol	16.38	0.94
10	*D*-carvone	17.08	0.64
11	Bornyl acetate	18.27	8.39
12	Myrtenyl acetate	19.30	3.77
13	*α*-cadino	23.72	0.64
14	Cubebol	23.81	1.63
15	∆-cadinene	24.36	0.78
16	Cyclohexene, 1,3-diisopropenyl-6-methyl	25.12	1.45
17	cis-.α.-copaene-8-ol	25.60	1.56
18	Caryophyllene oxide	26.21	0.89
19	Menthol	26.39	0.92
20	Viridiflorol	26.64	8.54
21	Acorenone B	26.75	1.03
22	Ledol	26.90	1.80
23	Humulane-1,6-dien-3-ol	27.04	4.56
24	Cedr-9-ene	27.52	1.35
25	*τ*-muurolol	27.95	2.7
26	Longiverbenone	28.78	1.32
27	β-copaen-4-ol	29.47	0.72
28	*Ɣ*-1-cadinene aldehyde	32.99	10.61

**Table 2 plants-12-01413-t002:** Chemical composition of LSEO_B_.

Peak Number	Compound	Retention Time	Area
1	*L*-fenchone	12.56	4.03
2	Linalool	12.95	2.48
3	Camphor	14.50	22.29
4	Pinocarvone	14.79	0.27
5	Borneol	15.14	5.15
6	*p*-menth-1-en-4-ol	15.33	1.93
7	Benzenemethanol, 4-(1-methylethyl)	15.54	2.21
8	Myrtenal	15.76	0.96
9	2-pinen-10-ol	15.82	0.54
10	Verbenone	16.14	2.66
11	2-cyclohexen-1-ol	16.42	0.96
12	D-carvone	17.10	0.41
13	Bornyl acetate	18.25	1.71
14	β-selinene	23.68	0.67
15	Myrtenyl acetate	24.04	0.46
16	cis-calamenene	24.43	2.79
17	Selina-3,7(11)-diene	25.08	2.00
18	Myrtenyl 2-methyl butyrate	25.28	0.41
19	Germacrene D-4-ol	25.48	0.5
20	1,3,3-trimethyl-2-(2-methylcyclopropyl)-1-cyclohexene	26.42	0.55
21	Eremophila ketone	26.60	1.13
22	2-octenoic acid	27.05	0.43
23	Cubebol	27.43	22.68
24	Aromadendrane-4,10-diol	27.54	0.55
25	*τ*-cadinol	27.97	2.63
26	Trans-valerenyl acetate	28.14	0.29
27	*τ*-muurolol	28.33	0.06
28	Silphiperfol-5-ene	28.66	3.27
29	Naphthalene, 1,6-dimethyl-4-(1-methylethyl)	28.78	1.11
30	Muurol-5-en-4-one <cis-14-nor->	29.20	4.21
31	*δ*-tridecalactone	29.54	0.44
32	1-naphthalenepropanol	29.80	2.94
33	Androstane-17,19-diol	30.53	0.29
34	Caryophyllene oxide	31.48	0.34
35	Neoisolongifolene	31.99	1.41
36	5-(7a-isopropenyl-4,5-dimethyl-octahydroinden-4-yl)-3-methyl-pent-2-en-1-ol	32.09	0.42
37	Longifolenaldehyde	32.53	0.58
38	Corymbolone	32.66	0.6
39	Myrtenyl acetate	35.81	0.83
40	Widdrol hydroxyether	36.06	0.31

**Table 3 plants-12-01413-t003:** Chemical composition of LSEO_K_.

Peak Number	Compound	Retention Time	Area
1	*L*-fenchone	12.55	1.88
2	Linalool	12.91	3.02
3	Camphor	14.43	13.39
4	Borneol	15.12	3.26
5	3-adamantan-1-yl-butan-2-one	15.33	4.39
6	Benzenemethanol, 4-(1-methylethyl)	15.60	3.00
7	2-pinen-10-ol	16.17	2.47
8	2-pinen-4-one	16.23	0.6
9	2-cyclohexen-1-ol	16.45	1.10
10	Verbenone	18.92	1.76
11	β-selinene	23.69	1.24
12	cis-calamenene	24.44	2.65
13	Selina-3,7(11)-diene	25.09	4.5
14	1,3,3-trimethyl-2-(2-methyl-cyclopropyl)-cyclohexene	26.43	0.65
15	Arctiol	27.07	1.20
16	Cubebol	27.40	16.07
17	*τ*-cadinol	27.96	2.08
18	*τ*-muurolol	28.54	18.44
19	Cedr-8(15)-en-9-ol	28.71	1.96
20	Naphthalene, 1,6-dimethyl-4-(1-methylethyl)	28.84	1.61
21	Muurol-5-en-4-one (cis-14-nor-)	29.29	6.84
22	*δ*-tridecalactone	29.58	1.74
23	1-naphthalenepropanol	29.81	2.10
24	2(3H)-naphthalenone	30.68	0.61
25	Caryophyllene oxide	31.51	0.57
26	Neoisolongifolene	32.00	0.81
27	Myrtenyl acetate	35.82	1.29
28	Methyl 5,9-docosadienoate	36.09	0.86

**Table 4 plants-12-01413-t004:** Antibacterial activity of LSEO_A_, LSEO_B_, and LSEO_K_ determined by disc diffusion method and their minimum inhibitory concentration (MIC) and minimum bactericidal concentration (MBC) (mg/mL).

Strains	LSEO_A_			LSEO_K_			LSEO_B_		
	DIZ *	MIC	MBC	DIZ	MIC	MBC	DIZ	MIC	MBC
*S. aureus*	6 ± 0.00 ^a^	NT	NT	6 ± 0.00 ^a^	NT	NT	7.66 ± 0.57 ^a^	NT	NT
*B. subtilis*	25 ± 4.35 ^a^	25	>50	21.66 ± 2.08 ^a^	6.25	>50	20.66 ± 1.15 ^a^	6.25	>50
*P. aeruginosa*	13.33 ± 1.15 ^a^	NT	NT	19 ± 1.00 ^b^	NT	NT	15.66 ± 0.57 ^a^	NT	NT
*P. mirabilis*	18.66 ± 1.15 ^a^	>50	>50	22.66 ± 0.57 ^b^	12.25	>50	20 ± 1.00 ^a^	12.5	25
*E. coli*	6 ± 0.00 ^a^	12.5	>50	10.66 ± 0.57 ^b^	3.12	>50	10 ± 0.00 ^b^	3.12	>50

* The diameter of the inhibition zones (mm), including diameter of disc 6 mm, are given as mean ± SD of triplicate experiments; DIZ: Diameter Inhibition Zones; NT: not tested; within each line, different letters (^a,b^) indicate significant differences (*p* < 0.05).

**Table 5 plants-12-01413-t005:** Molecular docking energy affinities of both receptors (pdb:6lu7 and pdb:6vsb) using Surflex-Dock and UCSF Chimera software.

Compounds	Surflex-Dock		UCSF Chimera	
	6lu7	6vsb	6lu7	6vsb
Cubebol	3.12	3.37	−5.5	−5.6
Bornyl acetate	3.92	3.55	−5.4	−5.3
L-fenchone	2.56	2.54	−4.2	−4.8
τ-muurolol	2.94	4.07	−5.3	−4.6
Viridiflorol	2.46	2.53	−5.5	−4.7
Camphor	2.60	3.26	−4.4	−4.4
Chloroquine	3.6	3.2	−5.7	−5.3

**Table 6 plants-12-01413-t006:** Physicochemical parameters (Lipinski Rule of Five) of cubebol and bornyl acetate compounds.

	MW	LogP	Rotatable Bonds	Donors	Acceptors	Surface
Cubebol	222.372	3.46	1	1	1	99.62
Bornyl acetate	196.29	2.76	1	2	0	86.01

**Table 7 plants-12-01413-t007:** Pharmacokinetic (ADMET) properties of cubebol and bornyl acetate compounds.

	Absorption			Distribution and Metabolism CYP450						Excretion and Toxicity		
	Skin Permeability	Intestinal Absorption	CaCO_2_ Permeability	3A4 Substrate	3A4 Inhibitor	6D6 Substrate	6D6 Inhibitor	VDss	BBB	Total Clearance	AMES	Hepatotoxicity
Cubebol	−2.17	94.94	1.32	yes	no	no	no	0.45	0.66	0.88	no	no
Bornyl acetate	−2.23	95.36	1.85	no	no	no	no	0.30	0.55	1.03	no	no

## Data Availability

Not applicable.

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
