# Peer review of "Chemical Profiling and Biological Properties of Essential Oils of Lavandula stoechas L. Collected from Three Moroccan Sites: In Vitro and In Silico Investigations"

_plants, 2023, doi:10.3390/plants12061413_

Round 1
Reviewer 1 Report
¾ The manuscript "Chemical profiling and biological properties of essential oils of Lavandula stoechas L. collected from three Moroccan sites: in vitro and in silico investigations" is a complex and interesting study, but it requires some improvements to make it more clearly presented. Even in the Abstract, unexplained abbreviations appear: LSEOA, LSEOK, and LSEOB, which are essential to understand the results and continue throughout the entire presentation. I would ask you to fix this, starting with the Abstract, Introduction, Materials, etc.
¾ Complete the working methods with data about the studied samples (pure or dissolved in alcohol Lavandula stoechas essential oil (LSEOs)? in the study of activities: antioxidants, anti-candida, Allelopathic.
¾ The formulas are not clear.
¾ Please improve the presentation of the species in terms of chemical composition.
¾ LSEOS and LSEOs - are they different?
¾ Conclusions should be made taking into account the results obtained comparatively.
Author Response
Response to Reviewer 1
Comment
The manuscript "Chemical profiling and biological properties of essential oils of Lavandula stoechas L. collected from three Moroccan sites: in vitro and in silico investigations" is a complex and interesting study, but it requires some improvements to make it more clearly presented.
Response: we would like to thank you for your valuable comments concerning our manuscript. The comments were very helpful for revising and improving our paper, as well as an important guiding significance to our research. We carefully studied the comments and have made correction which we hope meet with your approval. Revised portions are marked in yellow in the manuscript.
Comment
Even in the Abstract, unexplained abbreviations appear: LSEOA, LSEOK, and LSEOB, which are essential to understand the results and continue throughout the entire presentation. I would ask you to fix this, starting with the Abstract, Introduction, Materials, etc.
Response: all abbreviations were checked, unified, and defined from the first appearance of each abbreviated word throughout.
Comment
Complete the working methods with data about the studied samples (pure or dissolved in alcohol Lavandula stoechas essential oil (LSEOs)? in the study of activities: antioxidants, anti-candida, Allelopathic.
Response: checked and added; as well as different solvents used for essential oils concentration preparations were added to the manuscript.
Comment
The formulas are not clear.
Response: checked and clarified. In addition, we report that your remark allowed us to detect an error concerning "allelopathic activity". Indeed, we found that we studied the anti-proliferative effect of oils against L. sativum seeds and not their anti-germination effect; for this reason, and after discussion, the authors have decided to delete this part from manuscript.
Comment
Please improve the presentation of the species in terms of chemical composition.
Response: checked and improved.
Comment
LSEOS and LSEOs - are they different?
Response: the meaning is the same; we put the same abbreviation in the revised paper.
Comment
Conclusions should be made taking into account the results obtained comparatively.
Response: checked and improved according to the comments.
Reviewer 2 Report
The work of Benali and colleagues concerning the description of the chemical composition and the evaluation of the biological activity in terms of antioxidant activity, biological activity and allelopathic effects is interesting and well written but has important gaps in the chemistry section that need to be filled before considering the manuscript suitable for publication.
-In table 1 and 2, the authors report the retention times. These should be removed and replaced with the linear retention indices calculated according to Van den Dool and Kratz which are necessary for confirmation of identification of the listed compounds.
-Furthermore, alongside the calculated LRIs, those reported in the literature must also be inserted for a direct comparison so as to verify the accuracy of the identification.
-I consider it necessary to remove figures 1, 2 and 3 as the structures of those molecules are more than known so it is not relevant to include them in the paper.
-The conclusions section is too generic; I recommend adding something about the results obtained.
Author Response
Response to Reviewer 2
Comment
The work of Benali and colleagues concerning the description of the chemical composition and the evaluation of the biological activity in terms of antioxidant activity, biological activity and allelopathic effects is interesting and well written but has important gaps in the chemistry section that need to be filled before considering the manuscript suitable for publication.
Response: we would like to thank you for your important recommendations. Your comments certainly improved the quality of our manuscript. We carefully studied your suggestions and have made correction which we hope meet with your approval. Revised portions are marked in yellow in the manuscript.
Comment
In table 1 and 2, the authors report the retention times. These should be removed and replaced with the linear retention indices calculated according to Van den Dool and Kratz which are necessary for confirmation of identification of the listed compounds.
-Furthermore, alongside the calculated LRIs, those reported in the literature must also be inserted for a direct comparison so as to verify the accuracy of the identification.
Response: we agree with your opinion regarding the importance of retention index. But in our case we don't have the n-alkanes series to calculate the retention index of each compound and compare it with the literature according to the type of column used; therefore, the identification of each compound was based on its mass spectra (MS) and by computational matching with standard reference databases. In addition, the chemical composition was analyzed using GC-MS-MS attached to a triple quadrupole tandem mass spectrometer which is considered the best-in-class system for optimal sensitivity and quantification of targeted compounds. On the other hand, we have identified the compounds in the same way in our previous studies published in Plants and Molecules journals; please see the following links:
https://doi.org/10.3390/molecules27165157
https://doi.org/10.3390/plants11172226
Comment
I consider it necessary to remove figures 1, 2 and 3 as the structures of those molecules are more than known so it is not relevant to include them in the paper.
Response: thank you for your suggestion; the figures 1, 2, and 3 were removed.
Comment
The conclusions section is too generic; I recommend adding something about the results obtained.
Response: the section was checked and improved according to the comments.
Round 2
Reviewer 1 Report
The authors have made many additions and improvements.
Reviewer 2 Report
The authors proceeded to make the requested changes and adequately answered the questions.
I therefore believe that the work may be accepted for publication in its current revised form.